# Evaluation of the severity of nonalcoholic fatty liver disease through analysis of serum exosomal miRNA expression

**Jeong-An Gim[1]☯, Soo Min Bang[2]☯, Young-Sun Lee📵[2]‡\*, Yoonseok Lee[2], Sun Young Yim[2], Young Kul Jung[2], Hayeon Kim[3], Baek-Hui Kim[3], Ji Hoon Kim[2], Yeon Seok Seo[2], Hyung Joon Yim[2], Jong Eun Yeon📵[2]‡\*, Soon Ho Um[2], Kwan Soo Byun[2]**

**1** Medical Science Research Center, Korea University College of Medicine, Seoul, South Korea,
**2** Department of Internal Medicine, Korea University College of Medicine, Seoul, South Korea, **3** Department of Pathology, Korea University Guro Hospital, Korea University College of Medicine, Seoul, South Korea

☯ These authors contributed equally to this work.
‡ These authors also contributed equally to this work.
\* lys810@hanmail.net (Y-SL); jeyyeon@hotmail.com (JEY)

**Data Availability Statement:** All miRNA expression and the expression of the

## Abstract

Noninvasive techniques for evaluating the severity of nonalcoholic fatty liver disease (NAFLD) have shown limited diagnostic performance. MicroRNAs (miRNAs) are useful biomarkers for diagnosing and monitoring the progression and treatment response to several diseases. Here, we evaluated whether serum exosomal miRNAs could be used for the diagnosis and prognosis of NAFLD severity. Exosomal miRNAs were isolated from the sera of 41 patients with NAFLD (diagnosed using liver biopsy) for microarray profiling. The degree of NAFLD severity was determined using inflammation, steatosis, and ballooning scores and the NAFLD activity score (NAS). Correlations between miRNA expression, clinical and biochemical parameters, and mRNA expression were analyzed. Overall, 25, 11, 13, and 14 miRNAs correlated with the inflammation score, steatosis score, ballooning score, and NAS, respectively, with 33 significant correlations observed between 27 miRNAs and six clinical variables. Eight miRNAs (let-7b-5p, miR-378h, -1184, -3613-3p, -877-5p, -602, -133b, and 509-3p) showed anticorrelated patterns with the corresponding mRNA expression. In fibrosis, 52 and 30 interactions corresponding to high miRNA-low mRNA and low miRNA-high mRNA expression, respectively, were observed. The present results therefore suggest that serum exosomal miRNAs can be used to evaluate NAFLD severity and identify potential targets for NAFLD treatment.

## Introduction

Nonalcoholic fatty liver disease (NAFLD) comprises a broad spectrum of diseases, including nonalcoholic fatty liver, nonalcoholic steatohepatitis (NASH), cirrhosis, and hepatocellular carcinoma (HCC) [1]. NASH is a progressive disease that results in inflammation and hepatocyte injury, whereas nonalcoholic fatty liver or simple steatosis is a benign condition. To

corresponding target mRNA or gene files are available from the GSE89632 dataset.

**Funding:** This study was supported by a National Research Foundation of Korea grant from the Korean government (the Ministry of Education, Science and Technology 2019M3E5D1A01068997 and 2018R1A2B2006183). Research Supporting Program of the Korean Association for the Study of the Liver and the Korean Liver Foundation (KASLKLF2020-04).

**Competing interests:** The authors have declared that no competing interests exist.

evaluate the severity of NAFLD, the NASH Clinical Research Network developed the NAFLD activity score (NAS), which is a composite score of steatosis, inflammation, and hepatocyte ballooning [2]. Fibrosis, another important histological feature, is the most relevant factor associated with the prognosis of NAFLD [3]. Liver biopsy is the gold standard for the diagnosis of NASH and evaluation of NAFLD disease severity through analysis of histological findings [4]. However, liver biopsy has certain disadvantages, such as sampling error and inter-/intra-observer variation, and there is a risk of complications such as pain, infection, and bleeding [5]. Given the remarkable increase in global NAFLD prevalence in recent years, liver biopsies cannot be performed in all patients [6,7]. Therefore, noninvasive evaluation of NAFLD severity has been developed using serological and imaging biomarkers [8–10]. However, the clinical application of such noninvasive biomarkers has several disadvantages, such as high cost, availability, low accuracy and reliability, and limited validation. In addition to conventional radiological, serological, and pathological markers, exosomes and microRNAs (miRNAs) are promising biomarkers for the diagnosis and prognosis of NAFLD [11,12]. Exosomes contain several types of cargo molecules for cell signaling, such as proteins; lipids; and nucleic acids, including mRNAs, miRNAs, and other non-coding RNAs [13]. Among the non-coding RNAs, miRNAs regulate epigenetic gene expression by binding and suppressing their target mRNAs [14]. Therefore, miRNAs are involved in the progression of several diseases and have been studied as potential therapeutic targets [15]. Many miRNAs are involved in the progression of NAFLD and are emerging as novel biomarkers for distinguishing the different stages of NAFLD disease severity [16,17]. A recent meta-analysis identified several miRNAs as potential biomarkers of NAFLD and NASH, including miR-34a, miR-122, and miR-192 [18].

In the present study, we conducted a microarray analysis of serum exosomal miRNAs in patients with NAFLD diagnosed by biopsy. Following bioinformatic analysis, we observed important implications of exosomal miRNAs, which reflected disease severity, such as the grade of NAS score or degree of liver fibrosis. We also demonstrated a correlation between miRNAs and both clinical and biochemical parameters. Using a publicly available dataset, we observed miRNA–mRNA interactions that may help elucidate future targets for improvements in diagnostic and therapeutic performance.

## Materials and methods

### Study population

A total of 41 patients with NAFLD were included in this study. The inclusion criterion was the confirmation of NAFLD with liver biopsy. The exclusion criteria were as follows: 1) patients with other chronic liver diseases, including chronic viral hepatitis B and C and autoimmune liver disease, 2) excessive alcoholic consumption (men > 30 g/day and women > 20 g/day), 3) patients with decompensated liver cirrhosis, 4) patients with other severe systemic disease, and 5) pregnant women. Past medical history and clinical parameters were recorded, and biochemical analyses were performed during the admission period for liver biopsy. Biochemical analyses included tests for determination of hemoglobin (Hb), white blood cell (WBC), and platelet (PLT) levels; aspartate transaminase (AST), alanine transaminase (ALT), alkaline phosphatase (ALP), and gamma glutamyl (GGT) activities; total bilirubin and total albumin levels; prothrombin time (PT); and blood urea nitrogen (BUN), creatinine (Cr), and C-reactive protein (CRP) levels. This study was approved by the Institutional Review Board of Korea University Guro Hospital (2016GR0302). All participants provided written informed consent and agreed to provide their sera and clinical data for this study. This study was conducted in accordance with the Declaration of Helsinki, and all participating authors reviewed and approved the final version of the manuscript.

## Histopathological evaluation

Two specimens of liver tissue were obtained via a percutaneous liver biopsy. Following fixation, the sections were embedded in paraffin blocks and stained with hematoxylin and eosin and Masson's trichrome stain. Two experienced pathologists (HK and B-HK) evaluated each sample using the NASH Clinical Research Network histological scoring system [2]. NAS was calculated as the sum of scores for steatosis (0–3), inflammation (0–3), and hepatocyte ballooning (0–2). Fibrosis was classified into stages 0–4.

## Serum collection and exosomal isolation

After serum collection from patients with NAFLD and controls, sera were stored at -80˚C and thawed just before exosome isolation. We isolated exosomes from sera using an exosome isolation kit (ExoQuick Plus, Systemic Biosciences, CA, USA) according to the manufacturer's protocol. Sera (1 mL) were centrifuged at $3000 \times g$ for 15 min to remove cell and cell debris. Exoquick solution (63 μL) was added to the sera and incubated for 1 h at 4˚C. Mixtures were centrifuged at $1500 \times g$ for 30 min, and pellets were collected. The exosome pellets were resuspended in 150 μL of phosphate-buffered saline.

## Western blotting

Exosomal proteins were isolated using the M-PER buffer. Twenty micrograms of exosomes was used for immunoblotting using CD9 and CD63 antibodies (Systemic Biosciences, CA, USA). Protein separation was performed by electrophoresis using 10% sodium dodecyl sulfate-polyacrylamide gels, and the proteins were transferred to nitrocellulose membranes. Protein bands were visualized using enhanced chemiluminescence (Perkin Elmer, Waltham, MA, USA).

## RNA extraction and microarray analysis

RNA extraction from exosomes was performed in accordance with a previously published protocol [19]. Microarray analysis was performed using an Affymetrix GeneChip™ miRNA 4.0 array, and image and signal data were extracted using GeneChip™ Scanner 3000DX and Transcriptome Analysis Console 4.0. Normalization of each type of data was performed using the robust multichip average and detection above background algorithms in the Affymetrix Expression Console software. Normalized data are presented as a matrix, converted to an R data frame, and then subjected to statistical analyses. Analyses have been performed with R statistics v.3.6.1.

## Identification of differentially expressed miRNAs for each score and total NAS

We used ANOVA and ANOVA functions in R to identify differentially expressed miRNAs among each of the three groups, based on the inflammation score (0/1, 2, and 3), steatosis score (0/1, 2, and 3), ballooning score (0, 1, and 2), and total NAS (1–3, 4/5, and 6–8).

The heatmap and hierarchical clustering plot were obtained using the package "pheatmap" in R. The average expression of miRNAs in each group was calculated by the "rowMeans" function in R. The correlation between miRNA expression and clinical parameters of patients was evaluated using the package "pingouin" and the "corr" function in Python. The Circos plot was constructed using the package "circlize" in R.

### Assessing miRNA and mRNA interactions

To study the interaction between each miRNA expression and the expression of the corresponding target mRNA or gene, we used the publicly available GSE89632 dataset. In a previous study, the GSE89632 dataset was selected because it contains both fibrosis and NAS score information [20]. In the present study, the results for 41 patients were compared with those for 24 healthy controls, 20 patients with simple steatosis, and 19 patients with NASH. Gene expression profiles were confirmed using the Illumina HumanHT-12 WG-DASL V4.0 R2 expression beadchip.

### miRNA–mRNA interaction networks

For the prediction of targets modulated by miRNAs, we used the miRNA Data Integration Portal bioinformatics tool (http://ophid.utoronto.ca/mirDIP/). Cytoscape (version 3.8.0) was used to incorporate the identified miRNAs and target genes into the interaction network.

## Results

### Baseline characteristics

Histopathological, demographic, and laboratory findings and observations of 41 patients diagnosed with NAFLD based on histological examination are presented in Table 1. Twenty-two (53.7%) patients had NASH, as demonstrated by hepatocyte ballooning, and 16 patients had advanced fibrosis. The NAS score was distributed between 2 and 8 points, and 17 patients had a score of 5 or higher. Fibrosis stages of 0, 1, 2, 3, and 4 were observed in 8, 10, 7, 12, and 4 patients, respectively. Advanced stage fibrosis was observed in 16 (39.0%) patients. The patients were divided into groups with low, medium, and high NAS scores. Representative histopathological images of each group are shown in S1 Fig. The low NAS group showed mild steatosis and inflammation without hepatocyte ballooning or fibrosis, whereas the high NAS group showed severe steatosis and inflammation with ballooning and advanced fibrosis. The prevalence of diabetes/impaired fasting glucose, hypertension, and dyslipidemia was 68.3%, 46.3%, and 31.7%, respectively. The median age was 55 years, and the patients were predominantly women (68.3%). The median body mass index (BMI) was 29.94 kg/m$^2$.

### Analysis of expression patterns based on the degree of inflammation, steatosis, ballooning, and total NAS

Isolated exosomes were identified with western blotting for CD9 and CD63 (S2 Fig). We analyzed the three groups and observed that 25, 11, and 13 miRNAs were correlated with inflammation, steatosis, and ballooning scores, respectively (Fig 1A–1C); 14 miRNAs were correlated with NAS, and 21 miRNAs with NASH with significant fibrosis (NASH + NAS $\geq$ 4 + fibrosis $\geq$ 2) (Fig 1D and 1E). Furthermore, 23, 13, and 15 miRNAs showed significant correlations with three histological features, including NASH yes/no, NAS $\geq$ 4 yes/no, and advanced fibrosis (stage 3 or 4) yes/no, respectively (S3 Fig). Expression data for heatmaps were provided in the form of a matrix (S1 Table). The miRNA expression patterns with low *p*-values were selected based on the analysis of variance (ANOVA) performed by comparing the three groups for each variable (inflammatory score, steatosis score, ballooning score, and NAS) (Fig 2).

Comparison was performed with five normal controls without liver disease and the high-level group among the four variables. Differentially expressed miRNAs between normal and high-level groups are presented (S4 Fig), and the matrix of their expression levels were provided (S1 Table).

**Table 1. Patient information.**

| Characteristic | Total patients, n = 41 |
|---|---|
| Histological findings | |
| Steatosis score, n (%) 0/1/2/3 | 0 (0)/24 (58.4)/8 (19.5)/9 (22) |
| Lobular inflammation score, n (%) 0/1/2/3 | 0 (0)/15 (36.6)/20 (48.8)/6 (14.6) |
| Ballooning score, n (%) 0/1/2 | 19 (46.3)/12 (29.3)/10 (24.4) |
| NAS score n (%) 2/3/4/5/6/7/8 | 7 (17.1)/8 (19.5%)/9 (22)/8 (19.5)/6 (14.6)/2 (4.9)/1 (2.4) |
| Fibrosis stage, n (%) 0/1/2/3/4 | 8 (19.5)/10 (24.4)/7 (17.1)/12 (29.3)/4 (9.8) |
| Demographics | |
| Age, median (IQR), years | 55 (43–63) |
| Male, n (%) | 13 (31.7) |
| Diabetes/Impaired Fasting Glycemia, n (%) | 28 (68.3) |
| HTN, n (%) | 19 (46.3) |
| Dyslipidemia, n (%) | 13 (31.7) |
| BMI, median (IQR), kg/m$^2$ | 29.94 (26.20–33.46) |
| Laboratory findings | |
| Hemoglobin, median (IQR), g/dL | 14.0 (13.5–14.8) |
| WBC, median (IQR), $\times 10^3$/μL | 6.60 (5.20–8.60) |
| PLT, median (IQR), $\times 10^3$/μL | 206 (169–239) |
| AST, median (IQR), IU/L | 61 (42–79) |
| ALT, median (IQR), IU/L | 80 (50–112) |
| ALP (IU/L), median (IQR), IU/L | 95 (80–113) |
| GGT (IU/L), median (IQR), IU/L | 76 (49–103) |
| Bilirubin, median (IQR), mg/dL | 0.54 (0.43–0.67) |
| Albumin, median (IQR), g/dL | 4.3 (4.1–4.4) |
| PT, median (IQR), INR | 1.01 (0.97–1.05) |
| BUN, median (IQR), mg/dL | 13.6 (11.7–15.1) |
| Creatinine, median (IQR), mg/dL | 0.64 (0.56–0.78) |
| CRP, median (IQR), mg/L | 2.41 (1.10–4.07) |

n, number of patients; IQR, interquartile range; HTN, hypertension; BMI, body mass index; WBC, white blood cell; PLT, platelet; AST, aspartate transaminase; ALT, alanine transaminase; ALP, alkaline phosphatase; GGT, gamma glutamyl; PT, prothrombin time; BUN, blood urea nitrogen; CRP, C-reactive protein.

## Correlation patterns of miRNAs with clinical parameters in patients with NAFLD

Given that correlations were observed between the expression of certain miRNAs and disease severity, we next evaluated the correlation of the miRNAs with clinical parameters. The following 16 continuous variables, including clinical demographic parameters and laboratory findings, were selected for this evaluation: age, BMI, Hb, WBC, and PLT levels; total albumin and bilirubin; AST, ALT, ALP, and GGT activities; PT; BUN, Cr, and CRP levels; and NAS score. miRNAs with a statistical power of 0.9 or more were selected. In 41 patients, correlations were found between the 147 expression patterns of 117 miRNAs and 15 variables (S2 Table). We found that age, albumin, BUN, CRP, Hb, GGT, and PT were negatively correlated with the expression of some miRNAs, whereas ALP, ALT, AST, BUN, Cr, CRP, Hb, GGT, PT, PLT, WBC, and NAS scores were positively correlated with the expression of other miRNAs. BUN, CRP, Hb, GGT, and PT were selected as variables with positively and negatively correlated

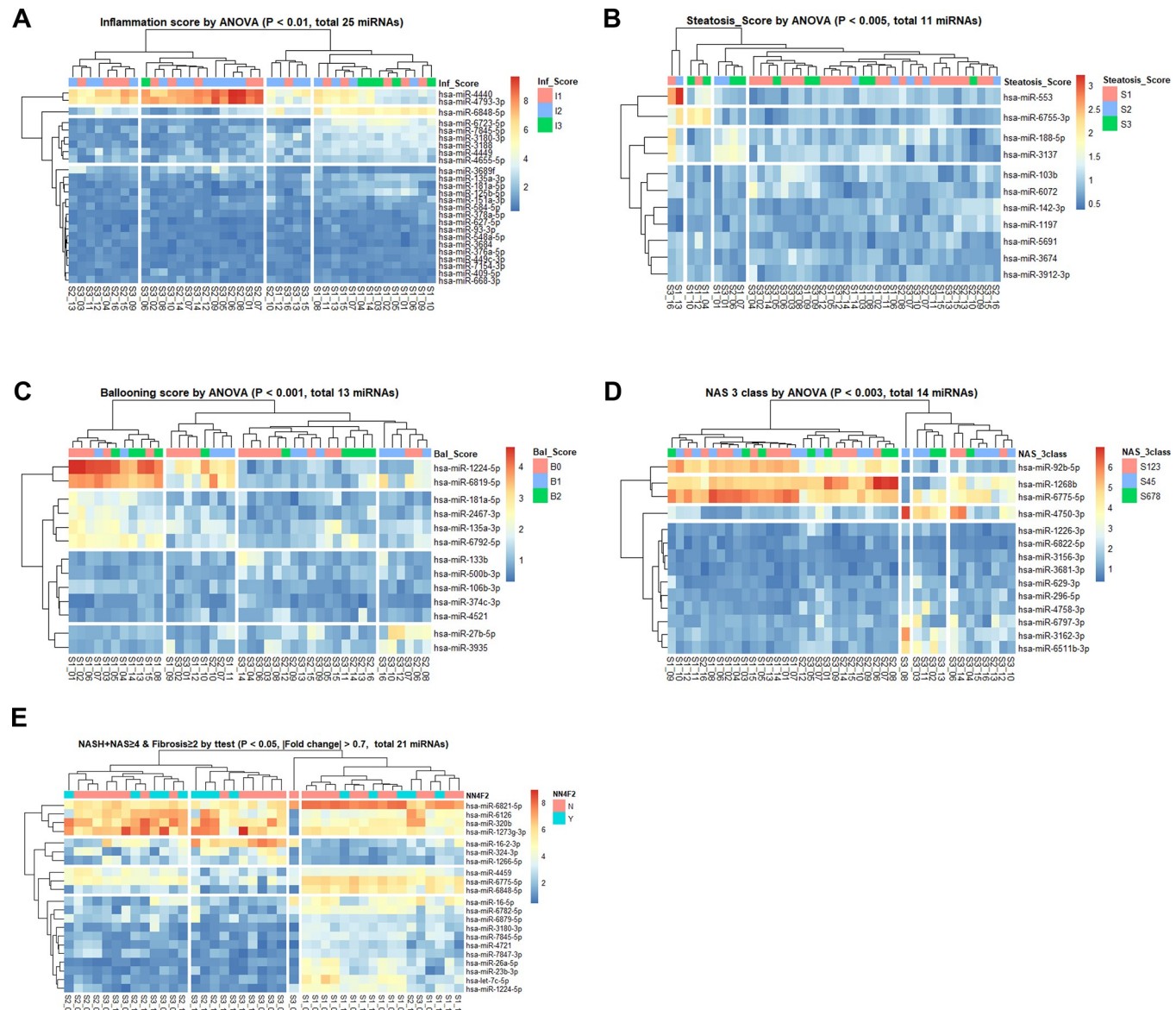

**Fig 1. Expression profile heatmaps of five variables.** Heatmaps of differentially expressed miRNAs (rows) from 41 patients (columns). At the top of each row, the scoring levels of 2 or 3 for each of the five variables are presented in the bar. Each row indicates identified miRNAs by the *t*-test or analysis of variance, and each column indicates a patient. Each row and column pair was clustered using the *k*-means clustering method with the package "pheatmap" in R and divided into four sections.

miRNAs. When we filtered r² based on a cut-off value of 0.35, we detected 33 correlation patterns between the expression of miRNAs and these six variables (Table 2). Although the expression of miR-4709-3p was negatively correlated with age, the expression of miR-133b and miR-8079 was positively correlated with ALP and ALT, respectively. AST and GGT levels were positively correlated with miR-4436a expression (S5 Fig). Total NAS was positively correlated with miR-7151-5p expression. When the expression level of hsa-miR-7151-5p exceeded 1.0, the NAS score was 6 or higher (S5 Fig).

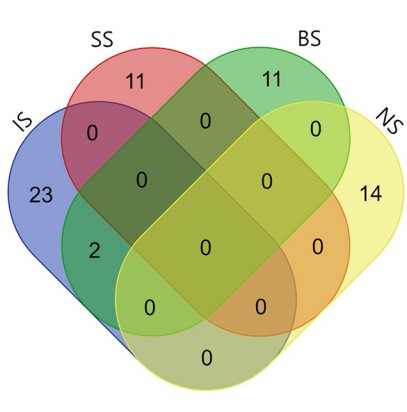

| Class | Method | List of identified miRNAs |
|---|---|---|
| Inflammatory score | ANOVA | hsa-miR-93-3p, hsa-miR-181a-5p, hsa-miR-125b-5p, hsa-miR-135a-3p, hsa-miR-376a-5p, hsa-miR-378a-5p, hsa-miR-151a-3p, hsa-miR-409-5p, hsa-miR-584-5p, hsa-miR-548a-5p, hsa-miR-627-5p, hsa-miR-668-3p, hsa-miR-449c-3p, hsa-miR-3180-3p, hsa-miR-3188, hsa-miR-3684, hsa-miR-4440, hsa-miR-4449, hsa-miR-3689f, hsa-miR-4655-5p, hsa-miR-4793-3p, hsa-miR-6723-5p, hsa-miR-6848-5p, hsa-miR-7154-3p, hsa-miR-7845-5p |
| Steatosis Score | ANOVA | hsa-miR-142-3p, hsa-miR-188-5p, hsa-miR-553, hsa-miR-1197, hsa-miR-103b, hsa-miR-3137, hsa-miR-3674, hsa-miR-3912-3p, hsa-miR-5691, hsa-miR-6072, hsa-miR-6755-3p |
| Ballooning Score | ANOVA | hsa-miR-181a-5p, hsa-miR-27b-5p, hsa-miR-135a-3p, hsa-miR-106b-3p, hsa-miR-133b, hsa-miR-1224-5p, hsa-miR-500b-3p, hsa-miR-3935, hsa-miR-374c-3p, hsa-miR-4521, hsa-miR-2467-3p, hsa-miR-6792-5p, hsa-miR-6819-5p |
| NAS | ANOVA | hsa-miR-296-5p, hsa-miR-92b-5p, hsa-miR-629-3p, hsa-miR-1226-3p, hsa-miR-3156-3p, hsa-miR-3162-3p, hsa-miR-3681-3p, hsa-miR-1268b, hsa-miR-4750-3p, hsa-miR-4758-3p, hsa-miR-6511b-3p, hsa-miR-6775-5p, hsa-miR-6797-3p, hsa-miR-6822-5p |

**Fig 2. Venn diagram of the miRNAs identified in each of the four variables by analysis of variance (ANOVA).** Each identified miRNA is listed in the table. IS, inflammatory score; SS, steatosis score; BS, ballooning score; NAS, nonalcoholic fatty liver disease (NAFLD) activity score.

## Comprehensive depiction of miRNA expression and its correlation with clinical features in patients with NAFLD using a Circos plot

We constructed a Circos plot to show the comprehensive correlation pattern of miRNA expression in each chromosome. Six variables, including fibrosis, NAS, presence or absence of diabetes, pathological parameters (inflammation, steatosis, and ballooning score), and biochemical parameters (ALT and PLT levels) were selected. Each variable was grouped into the categories of low or high scores or values (fibrosis stages 0–2 versus 3 or 4; NAS score <5 vs. ≥5; diabetes, yes versus no; inflammation score 1 versus 2 versus 3; steatosis score 1 versus 2 versus 3; ballooning score 0 versus 1 versus 2), and the correlation between miRNA expression and two biochemical parameters was analyzed (ALT and PLT) (Fig 3). Differences between the mean values of the two or three categories of datasets were obtained for the expression of 2,576 miRNAs, and the top 200 miRNAs were selected to construct the Circos plot. The top 200 selected miRNAs were located in the corresponding genomic regions in the Circos plot. miRNA expression was altered between the two or three groups. Among the clinical parameters, ALT and PLT were correlated with each group, as shown by the two internal peaks (Fig 3). When the miRNA expression of ALT and PLT according to the group was positively correlated, a yellow or red peak appeared, and a negative correlation was indicated by a green or sky-blue peak. The correlation of miRNA expression at each genomic location was confirmed by the height of the peak.

## miRNA–mRNA interaction networks

Co-expression networks between the correlated mRNAs and miRNAs were analyzed based on the miRNA expression data obtained in the present study and from the publicly available mRNA expression dataset GSE89632. This dataset was selected because it contains both fibrosis and NAS scores. miRNAs demonstrating significant alterations in their expression patterns associated with total NAS scores and fibrosis were selected. Using the same approach as in a previous study when analyzing the GSE89632 dataset [20], we identified 385 and 359 genes with high and low expression, respectively. Considering that the expression of miRNAs in the serum and liver tissue demonstrates a variable correlation, we matched two-by-two correlations between serum miRNA and mRNA expression patterns (higher miRNA–lower mRNA,

**Table 2. Correlations between miRNA expression and clinical parameters.**

| Variable | miRNA name | r | 95% Confidence interval | r² | p-value | power |
|---|---|---|---|---|---|---|
| Age | hsa-miR-4709-3p | -0.621 | [-0.78–0.39] | 0.385 | 1.49E-05 | 0.995 |
| Age | hsa-miR-4999-5p | -0.604 | [-0.77–0.36] | 0.365 | 2.91E-05 | 0.992 |
| ALP | hsa-miR-133b | 0.64 | [0.41 0.79] | 0.41 | 6.63E-06 | 0.997 |
| ALT | hsa-miR-8079 | 0.621 | [0.39 0.78] | 0.386 | 1.45E-05 | 0.995 |
| AST | hsa-miR-548ah-3p | 0.618 | [0.38 0.78] | 0.382 | 1.65E-05 | 0.994 |
| AST | hsa-miR-644a | 0.625 | [0.39 0.78] | 0.391 | 1.24E-05 | 0.995 |
| AST | hsa-miR-19a-3p | 0.632 | [0.4 0.79] | 0.399 | 9.38E-06 | 0.996 |
| AST | hsa-miR-577 | 0.638 | [0.41 0.79] | 0.407 | 7.22E-06 | 0.997 |
| AST | hsa-miR-559 | 0.654 | [0.43 0.8] | 0.428 | 3.50E-06 | 0.998 |
| AST | hsa-miR-4291 | 0.669 | [0.46 0.81] | 0.448 | 1.71E-06 | 0.999 |
| AST | hsa-miR-4436a | 0.703 | [0.5 0.83] | 0.494 | 3.05E-07 | 1 |
| GGT | hsa-miR-3202 | 0.594 | [0.35 0.76] | 0.353 | 4.25E-05 | 0.989 |
| GGT | hsa-miR-488-5p | 0.604 | [0.36 0.77] | 0.365 | 2.90E-05 | 0.992 |
| GGT | hsa-miR-659-5p | 0.605 | [0.36 0.77] | 0.366 | 2.84E-05 | 0.992 |
| GGT | hsa-miR-4512 | 0.606 | [0.37 0.77] | 0.367 | 2.72E-05 | 0.992 |
| GGT | hsa-miR-3690 | 0.622 | [0.39 0.78] | 0.387 | 1.40E-05 | 0.995 |
| GGT | hsa-miR-599 | 0.639 | [0.41 0.79] | 0.408 | 6.92E-06 | 0.997 |
| GGT | hsa-miR-3925-5p | 0.64 | [0.41 0.79] | 0.41 | 6.63E-06 | 0.997 |
| GGT | hsa-miR-19a-3p | 0.645 | [0.42 0.8] | 0.416 | 5.26E-06 | 0.998 |
| GGT | hsa-miR-577 | 0.65 | [0.43 0.8] | 0.422 | 4.29E-06 | 0.998 |
| GGT | hsa-miR-4650-5p | 0.657 | [0.44 0.8] | 0.431 | 3.12E-06 | 0.998 |
| GGT | hsa-miR-4694-3p | 0.66 | [0.44 0.8] | 0.435 | 2.73E-06 | 0.999 |
| GGT | hsa-miR-3157-5p | 0.66 | [0.44 0.8] | 0.436 | 2.62E-06 | 0.999 |
| GGT | hsa-miR-130b-5p | 0.661 | [0.44 0.81] | 0.437 | 2.50E-06 | 0.999 |
| GGT | hsa-miR-518f-5p | 0.682 | [0.47 0.82] | 0.465 | 9.07E-07 | 0.999 |
| GGT | hsa-miR-3660 | 0.683 | [0.47 0.82] | 0.466 | 8.76E-07 | 0.999 |
| GGT | hsa-miR-548ah-3p | 0.714 | [0.52 0.84] | 0.509 | 1.64E-07 | 1 |
| GGT | hsa-miR-4291 | 0.722 | [0.53 0.84] | 0.521 | 9.93E-08 | 1 |
| GGT | hsa-miR-2053 | 0.724 | [0.54 0.84] | 0.524 | 8.88E-08 | 1 |
| GGT | hsa-miR-644a | 0.743 | [0.56 0.85] | 0.552 | 2.71E-08 | 1 |
| GGT | hsa-miR-33b-3p | 0.765 | [0.6 0.87] | 0.585 | 5.88E-09 | 1 |
| GGT | hsa-miR-4436a | 0.829 | [0.7 0.91] | 0.687 | 2.21E-11 | 1 |
| NAS_Score | hsa-miR-7151-5p | 0.627 | [0.4 0.78] | 0.393 | 1.14E-05 | 0.996 |

ALP, alkaline phosphatase; ALT, alanine aminotransferase; AST, aspartate aminotransferase; GGT, gamma glutamyl transferase; NAFLD, nonalcoholic fatty liver disease; NAS, NAFLD activity score.

lower miRNA–higher mRNA, lower miRNA–lower mRNA, and higher miRNA–higher mRNA). With respect to fibrosis, we observed 52 interactions corresponding to high miRNA–low mRNA expression and 30 interactions corresponding to low miRNA–high mRNA expression using the mirDIP program [21]. Eight miRNAs, namely let-7b-5p and miR-378h, -1184, -3613-3p, -877-5p, -602, -133b, and 509-3p showed anticorrelated patterns with the corresponding mRNA expression. One of the downregulated miRNAs, has-let-7b-5p, was correlated with 30 mRNAs (Fig 4A). Five miRNAs were highly expressed in the high-fibrosis samples, of which 39 genes were downregulated (Fig 4B). With respect to the NAS score, 13 and 2 interactions were observed, corresponding to low miRNA–high mRNA (Fig 4C) and high miRNA–low mRNA expression patterns, respectively (Fig 4D). The eight miRNAs

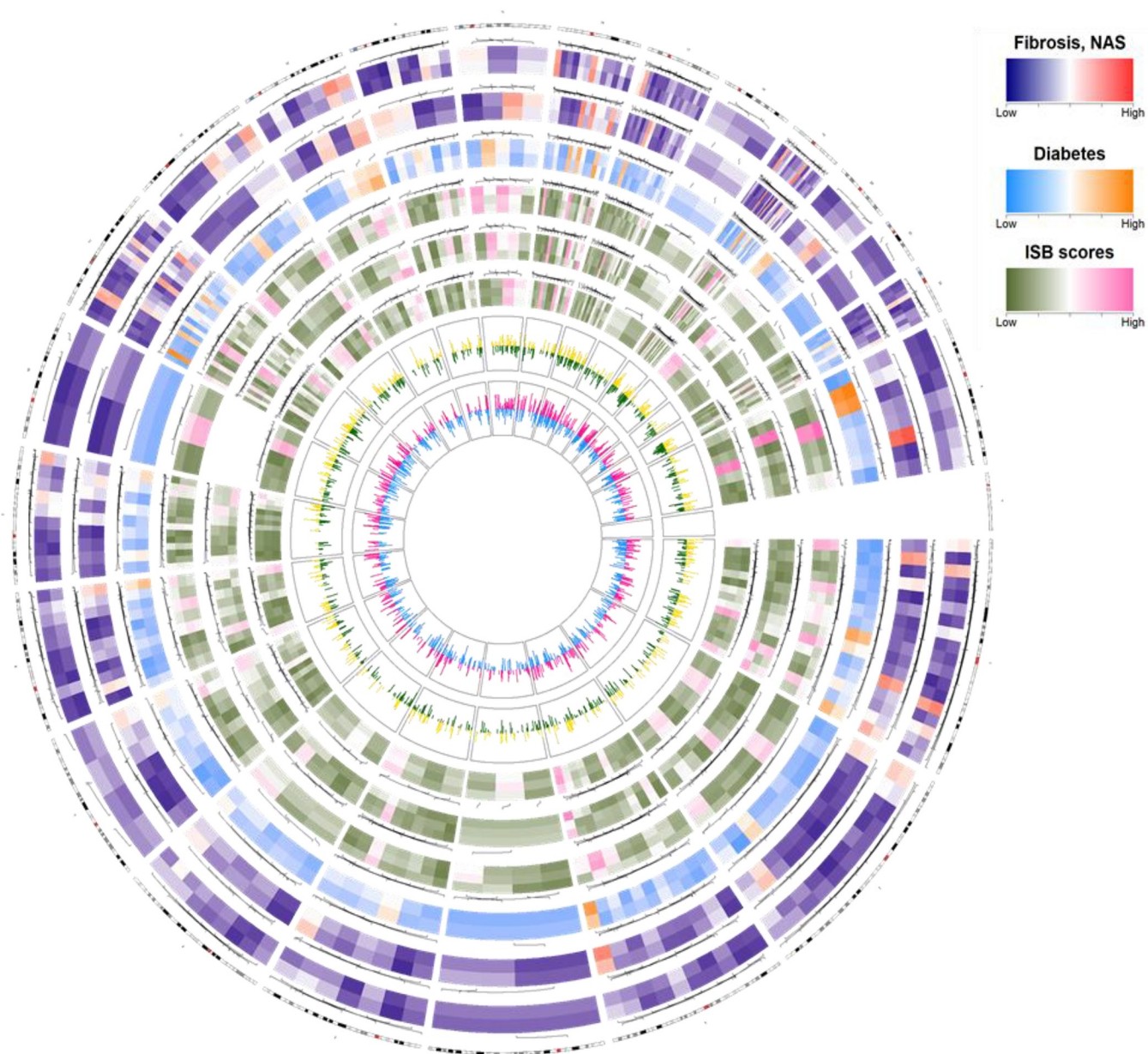

**Fig 3. Circos plot presenting the summary of miRNA expression and correlation patterns in 41 patients.** The Circos plot consists of eight tracks, containing six heatmaps for six variables (fibrosis, nonalcoholic fatty liver disease, diabetes, inflammation score, steatosis score, and ballooning score); the two peaks indicate the correlations between miRNAs and two clinical parameters (ALT, alanine aminotransferase; PLT, platelet). The six variables are represented by colors according to high (red, orange, and pink) or low (blue, sky blue, and green) expression. Three variables (inflammation score, steatosis score, and ballooning score) are represented as I, S, and B in the color key. Each variable was grouped into two or three categories of low or high scores or values (fibrosis stage, 0–2 versus 3 and 4; NAS < 5 versus ≥ 5; diabetics, yes versus no; inflammation score 1 versus 2 versus 3; steatosis score 1 versus 2 versus 3; ballooning score 0 versus 1 versus 2). Differences between the mean values of each dataset for the six variables were obtained. Differences in the mean values for each dataset were obtained from the expression levels of 2,576 miRNAs, and the top 200 miRNAs were plotted in the heatmap of the Circos plot. The positioning of each heatmap represents the genomic location of the miRNA, and the color represents the average expression level in patients belonging to each score of each variable. The outer yellow or red indicates a positive correlation between ALT or PLT and miRNA expression, and the inner green or sky blue indicates a negative correlation.

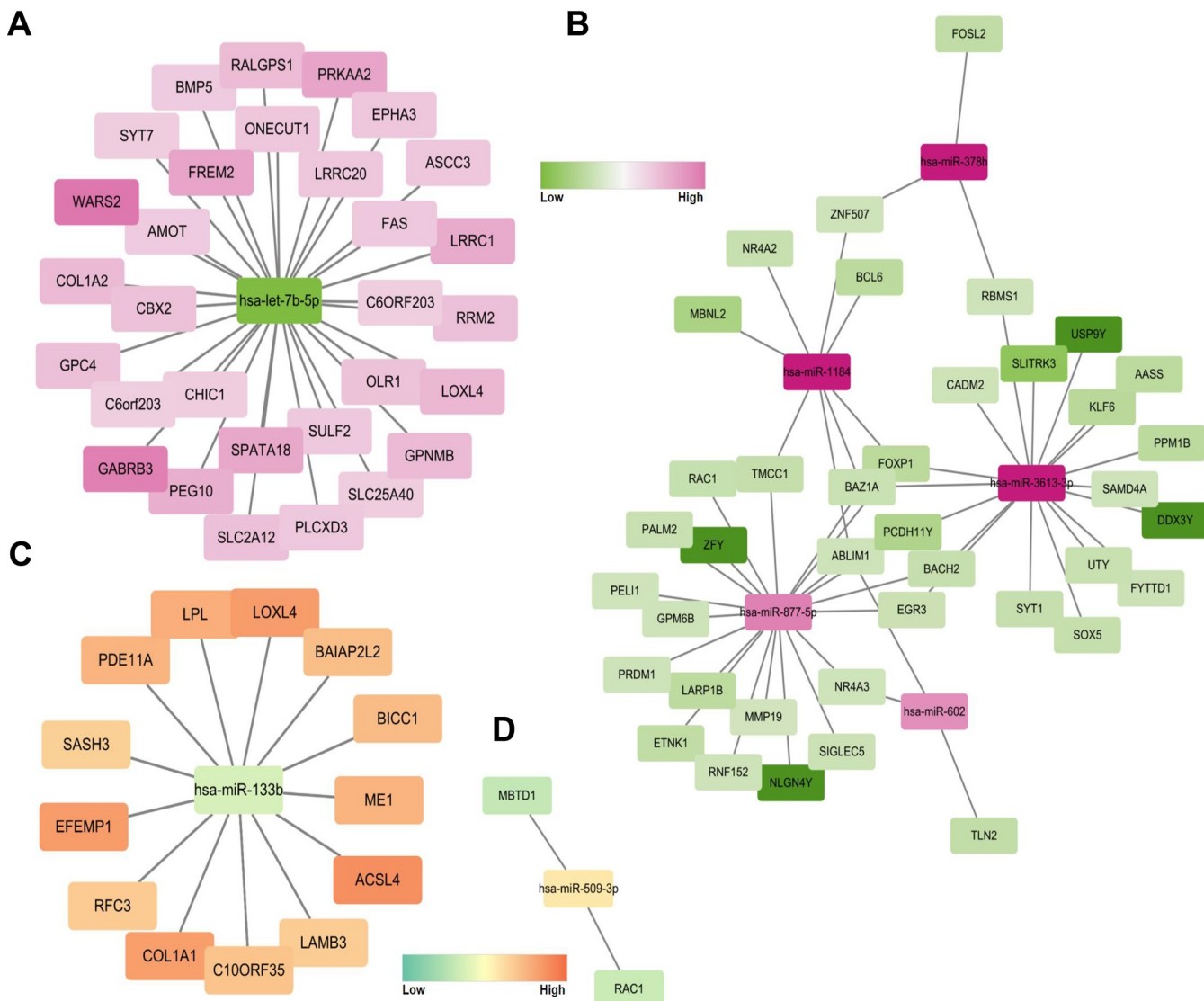

**Fig 4. Network analysis of mRNA–miRNA interactions with the GSE89632 dataset.** A network between a less expressed miRNA (our data) and a more expressed gene (GSE89632) (A), and between a more expressed miRNA (our data) and a less expressed gene (GSE89632) (B) in people with high fibrosis. The closer to green, the less expressed (that is, more likely to be affected by miRNA), and the closer to pink, the more expressed. A network between a less expressed miRNA (our data) and a more expressed gene (GSE89632) (C), and between a more expressed miRNA (our data) and a less expressed gene (GSE89632) (D) in people with high NAS. The closer to green, the less expressed (that is, more likely to be affected by miRNA), and the closer to orange, the more expressed.

showed little statistical difference according to three variables (diabetes, dyslipidemia, or sex) and can be interpreted as confounding variables (S6 Fig).

## Discussion

As NAFLD is a broad-spectrum disease with pathologies ranging from simple steatosis to cirrhosis or HCC [22], it is difficult to distinguish between the stages of NAFLD without liver biopsy, which is an invasive procedure. miRNAs are novel candidates as potential diagnostic and prognostic biomarkers of NAFLD because they are abundantly present in the liver and are associated with various types of liver diseases [23]. We observed that several miRNAs showed

significant differences in expression according to the severity of NAFLD and histological findings.

In patients with NAFLD diagnosed with NASH or advanced fibrosis (fibrosis stage 3 or 4), the prognosis is typically poor compared to that in patients with simple steatosis [3]. In patients with NAFLD, NASH represents a severe form of hepatic damage due to the recruitment of pro-inflammatory immune cells, which can eventually progress to cirrhosis and hepatocellular carcinoma [24]. Steatosis evaluation is important for the diagnosis of NAFLD, and changes in steatosis are considered indicators for improvement in clinical trials [25]. Liver biopsy is the gold standard for the diagnosis of NASH with advanced fibrosis or evaluation of steatosis in patients. However, it is impossible to perform a liver biopsy in all patients with NAFLD, which account for 25%–30% of the general population [26].

Therefore, biomarkers for the noninvasive evaluation of NAFLD, such as serological and imaging biomarkers, have been investigated [9]. Some biomarkers are based on a single parameter, whereas others have been developed as panels that combine several parameters [8]. Our study demonstrated that serum exosomal miRNAs showed significant differences according to the severity of NAFLD. NASH, an advanced form of NAFLD, is histologically characterized by inflammation, steatosis, and hepatocyte ballooning, and each histological feature is associated with different miRNAs. The diversity of miRNA expression profiles indicates that miRNAs are independently involved in steatosis, inflammation, and ballooning processes.

Although NAS is defined as the sum of steatosis, inflammation, and hepatocyte ballooning scores, on performing an ANOVA, different miRNA expression patterns were associated with each of these. However, in the Circos plot, the average miRNA expression level for each degree of the variables showed a similar pattern in each genomic region. Many miRNAs were observed corresponding to chromosomes 16, 17, and 19, which showed significant differences based on the score, and a few miRNAs were discovered corresponding to chromosomes 4 and 10. No miRNA was correlated with the steatosis score on chromosome 10. Interestingly, higher or lower correlation patterns between miRNA expression and two clinical parameters (ALT and PLT) were also detected in chromosomes 16, 17, and 19.

In addition, we identified correlations between laboratory data and the expression of serum exosomal miRNAs. Alterations in the levels of liver enzymes, such as AST, ALT, ALP, and GGT, are important not only for the differential diagnosis of liver disease but also for evaluating disease severity [27]. To date, differentially expressed miRNAs in liver injury status, such as oxidative stress or viral infection, have been detected [23,28–30]. It is necessary to comprehensively understand the relationship between miRNA and liver enzymes derived from biochemical analyses, as well as miRNA and mRNA derived from previous studies related to the liver injury status. However, no study has reported the relationship between liver enzymes and miRNAs. In the present study, we identified four miRNAs that were positively correlated with liver enzymes, namely miR-133b, miR-4436a, miR-4709-3p, and miR-8079.

Functional network analysis between mRNAs and miRNAs is useful for evaluating the course of various diseases [31,32]. We identified relationships between the miRNA expression patterns observed in the present study and the gene expression patterns demonstrated in the GSE89632 dataset. In this dataset, subjects are divided into healthy controls and patients with steatosis and NASH, demonstrating characteristics of fibrosis (stage), steatosis (%), lobular inflammation (severity), ballooning (score), and NAS [20]. The study methodology was similar to that used in the present study. Hence, we used this dataset for the comprehensive network analysis. Regarding hepatic fibrosis, we found that low expression of hsa-let-7b-5p was correlated with high expression of genes associated with liver fibrosis. The interaction between pro-fibrosis genes and hsa-let-7b-5p has been identified in interstitial pulmonary fibrosis and renal fibrosis [33,34]. *WARS2* and *GABRB3* were expressed at high levels. As for the NAS, low

expression of miR-133b was correlated with high expression of several genes, such as *LOXL4*, *ACSL4*, *COL1A1*, and *EFEMP1*. As miR-133b showed protective effects against allergic inflammation [35], its lower expression might increase hepatic inflammation. We attempted to link the miRNAs and mRNAs that have been previously studied in pathological liver injury such as NAFLD, NASH, and liver fibrosis [30,36–38]. Although it was difficult to identify genes directly related to liver injury in the interaction study, the miRNA-mRNA interaction presented in this study will serve as evidence to explain NAFLD and NASH well. Based on these observations, the interaction between miRNAs and mRNAs is believed to play an important role in the progression of NASH.

The scarcity of publicly available gene expression data contributes to the limitations of our study. To the best of our knowledge, studies with similar approaches that provide clinical parameters have not been performed previously, except for the GSE89632 study. The evidence demonstrated in our miRNA expression dataset needs to be validated in independent cohorts with larger sample sizes. The evidence and methods provided by us could be expanded to a comprehensive study of miRNAs and genes related to steatosis, fibrosis, ballooning, diabetes, and NAS. The small number of patients in each group for evaluating each variable was also a limitation. When ANOVA was performed with four variables, namely inflammation, steatosis, ballooning score, and NAS, it was difficult to ascertain a clear classification pattern according to the miRNAs describing each variable. Although the *p*-value indicating significance was different for each variable, a minimum of 11 and a maximum of 25 miRNAs explained the classification of each variable. These data need to be verified with larger patient cohorts in future studies. Although the main purpose of this study was to evaluate the severity of NAFLD, conducting the study without healthy controls also constitutes a limitation.

We identified differentially expressed miRNAs based on the groups of four variables with high, middle, and low values corresponding to the degree of severity (inflammation, steatosis, ballooning score, and NAS). Although NAS comprises three variables, different miRNAs were identified that are associated with each variable. Therefore, the pathways that explain each variable may differ. Comparative visualization was attempted with the construction of a Circos plot. miRNAs that were significantly different with respect to each variable showed similar expression patterns in similar genomic regions. The integration of miRNA and mRNA expression analyses and network analysis also enabled us to interpret the differentially regulated genes in NASH from the perspective of systems biology. Using our methodology, co-expression networks of miRNAs and mRNAs could help reveal the pathological pathways of NAFLD, as well as provide new insights into several biological pathways related to liver function. These pathways may also be used for the diagnosis of many liver diseases.

## Supporting information

**S1 Fig. Histopathological findings for representative patients in the low, medium, and high NAS groups.** Steatosis scores: 0 ($<$ 5%), 1 (5%–33%), 2 ($>$ 33%–66%), or 3 ($>$ 66%). Inflammation scores: 0 (no foci), 1 ($<$ 2 foci per 200× field), 2 (2–4 foci per 200× field), or 3 ($>$ 4 foci per 200× field). Hepatocyte ballooning scores: 0 (none), 1 (few), or 2 (many). Fibrosis stage: 0 (none), 1 (perisinusoidal or periportal), 2 (perisinusoidal and portal/periportal), 3 (bridging fibrosis), or 4 (cirrhosis). *NAS, nonalcoholic fatty liver disease (NAFLD) activity score; H&E, hematoxylin and eosin staining.
(TIF)

**S2 Fig. Identification of exosomes with western blotting.** Expression levels of CD9 (28 kDa) and CD63 (53 kDa) in four samples.
(TIF)

**S3 Fig. Expression profiles of three variables.** Heatmaps of differentially expressed miRNAs (rows) from 41 patients (columns) are displayed for NASH (A), NAS (B), and fibrosis (C). The classification of the variable for each patient at the top of the rows is displayed in the bar. Each row indicates the miRNAs identified by a *t*-test, and each column indicates a patient. Each row and column pair was clustered by the *k*-means clustering method using the package "pheatmap" in R, and divided into four sections. NASH, nonalcoholic steatohepatitis; NAS, nonalcoholic fatty liver disease (NAFLD) activity score.
(TIF)

**S4 Fig. Differentially expressed miRNAs between five normal controls and four patients with high scores.** The highest score among the four scores (steatosis score 3; n = 10, inflammation score 3; n = 6, ballooning score 1 or 2; n = 22, and NAS 6, 7, or 8; n = 10) and five normal controls were compared. (A) A total of 19 miRNAs with a difference in expression between five normal controls and 10 steatosis score 3 (S3; $p < 0.001$ and |fold change| > 1.8). (B) A total of 17 miRNAs with a difference in expression between five normal controls and six inflammation score 3 (I3; $p < 0.001$ and |fold change| > 1.5). (C) A total of 17 miRNAs with a difference in expression between five normal controls and 22 ballooning scores of 1 or 2 ($p < 0.001$ and |fold change| > 1.8). HB indicates a high ballooning score, considered as a score of 1 or 2. (D) A total of 17 miRNAs with a difference in expression between five normal controls and 10 NAS scores of 6, 7, or 8 (S678; $p < 0.001$ and |fold change| > 1.1). (E) Venn diagram and table by variable for each miRNA. S, I, B, and N indicate steatosis, inflammation, ballooning, and NAS (nonalcoholic fatty liver disease (NAFLD) activity score), respectively.
(TIF)

**S5 Fig. Correlation between miRNA expression levels and six clinical parameters.** In the scatter plot, the x-axis indicates the miRNA expression levels, and the y-axis indicates each clinical parameter. The lines and shades indicate the regression and confidence intervals, respectively.
(TIF)

**S6 Fig. Boxplot of significance and expression levels according to the confounding variables of eight miRNAs suggested to explain NAFLD in this study.** Eight miRNAs (let-7b-5p, miR-378h, -1184, -3613-3p, -877-5p, -602, -133b, and 509-3p) and three confounding variables (sex, diabetes, and dyslipidemia) were listed, and the expression levels are presented as boxplots with significance values (*p*-values). NAFLD, nonalcoholic fatty liver disease.
(TIF)

**S1 Table. Expression profiles are listed in 12 tables.** These datasets are provided as separate files.
(XLSX)

**S2 Table. Correlations between the 147 expression patterns of 117 miRNAs and 15 selected variables.** Each correlated miRNA with each variable is listed, and its genomic location is provided (miRbase Release 20). Correlations between variables and miRNA expression levels were estimated using the "pingouin" library in Python. Samples with "power" > 0.9, were selected. "n," "r," "CI95," "adj_r2," "p-val," and "BF10" represent "sample number," "correlation coefficient," "95% confidence interval," "adjusted r square," "*p*-value," and "Bayes factor of the alternative hypothesis," respectively. Power was calculated using the "1-type II error".
(DOCX)

## Author Contributions

**Conceptualization:** Jeong-An Gim, Young-Sun Lee, Jong Eun Yeon.

**Data curation:** Jeong-An Gim, Soo Min Bang, Young-Sun Lee, Yoonseok Lee, Sun Young Yim, Young Kul Jung, Hayeon Kim, Baek-Hui Kim.

**Formal analysis:** Soo Min Bang.

**Supervision:** Ji Hoon Kim, Yeon Seok Seo, Jong Eun Yeon.

**Visualization:** Jeong-An Gim.

**Writing – original draft:** Jeong-An Gim, Young-Sun Lee, Jong Eun Yeon.

**Writing – review & editing:** Hyung Joon Yim, Jong Eun Yeon, Soon Ho Um, Kwan Soo Byun.

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
