## [Decision Letter · Decision Letter 0]

24 Jun 2021

PONE-D-21-16287

Evaluation of the severity of nonalcoholic fatty liver disease by the analysis of serum exosomal miRNA expression

PLOS ONE

Dear Dr. Yeon,

Thank you for submitting your manuscript to PLOS ONE. After careful consideration, we feel that it has merit but does not fully meet PLOS ONE’s publication criteria as it currently stands. Therefore, we invite you to submit a revised version of the manuscript that addresses the points raised by the reviewer.

We look forward to receiving your revised manuscript.

Kind regards,

Matias A Avila, Ph.D.

Academic Editor

PLOS ONE

Journal Requirements:

Reviewers' comments:

Reviewer's Responses to Questions

**Comments to the Author**

1. Is the manuscript technically sound, and do the data support the conclusions?

Reviewer #1: Partly

2. Has the statistical analysis been performed appropriately and rigorously? 

Reviewer #1: I Don't Know

3. Have the authors made all data underlying the findings in their manuscript fully available?

Reviewer #1: Yes

4. Is the manuscript presented in an intelligible fashion and written in standard English?

Reviewer #1: Yes

5. Review Comments to the Author

Reviewer #1: In this study, Jong Eun Yeon et al., evaluated whether serum exosomal miRNAs could be used for the diagnosis and prognosis of NAFLD severity. They identified 25, 11, 13, and 14 miRNAs that correlated with inflammation, steatosis, ballooning and NAS, respectively. However, the study has limitations due to the small number of patients used in each group, and it would be important to validate these data with a different and larger patient cohort. Furthermore, as mentioned by authors in the discussion, the study has been conducted without healthy controls, which is another clear limitation.

Main

1) A comparison of the miRNA expression patterns of each group (inflammation score, steatosis score, ballooning score, and NAS) with healthy controls, is necessary to further confirm that serum exosomal miRNAs can be used to evaluate NAFLD severity.

2) Is there any correlation between the miRNAs identified with diabetes, dyslipidemia or gender?

3) The reference mentioned in the manuscript for the exosome isolation method is related with exosome isolation from cell cultures. ¿Could the authors describe the procedure of exosome isolation? Including volume of serum used, sample preservation, etc…

4) The exosomes isolated from the serum should be assessed by analyzing vesicle diameter by electron microscopy or by assessing common exosomal proteins by western blot.

5) A better description for the inclusion and exclusion criteria should be included, define alcohol consumption for women and men, viral-, autoimmune-, hemochromatosis- and drug-induced causes of liver disease, etc…

6) Can the authors be more specific with the classification used to classify the patients into groups with low, medium, and high NAS scores.

7) The supplementary figure 3 contain important information about the miR-RNAs identified, it would be useful to move this supplementary figure to main figures.

8) It should be also helpful to discuss the known functions of some of the MiR-RNAs identified, as for instance miR-125b-5p (PMID: 26924666), miR12245p (PMID: 30087537), miR-133b etc.

6. PLOS authors have the option to publish the peer review history of their article (what does this mean?). If published, this will include your full peer review and any attached files.

Reviewer #1: No

---

## [Author Response · Author response to Decision Letter 0]

19 Jul 2021

Dear Editor and Reviewer:

We are very grateful for your thoughtful and helpful comments on our manuscript titled “Evaluation of the severity of nonalcoholic fatty liver disease through analysis of serum exosomal miRNA expression” (Manuscript ID, PONE-D-21-16287). We have made a significant effort to incorporate all changes suggested by the reviewer and believe these have resulted in a significantly improved manuscript that we hope is now fit for publication in Plos One. The changes made are shown in red font in the revised manuscript.

Our point-by-point replies to the reviewer’s comments are provided below.

Looking forward to hearing from you,

Kind regards,

Jong Eun Yeon

Department of Internal Medicine, College of Medicine,

Korea University Guro Hospital,

148 Gurodong-ro, Guro-Gu, Seoul 08308, S. Korea

Phone/Fax: +82 2 2626 1030 / +82 2 2626 1038

jeyyeon@hotmail.com 

Reviewer 

1) A comparison of the miRNA expression patterns of each group (inflammation score, steatosis score, ballooning score, and NAS) with healthy controls, is necessary to further confirm that serum exosomal miRNAs can be used to evaluate NAFLD severity.

Response: We agree with the reviewer’s opinion and have used the results of five normal samples previously analyzed as controls. The highest score among the four scores (inflammation score, 3; n = 6, steatosis score, 3; n = 10, ballooning score, 1 or 2; n = 22, and NAS 6- 8; n = 10) and five normal controls were compared. The results are shown as a heatmap and presented in S3 Fig. Each matrix of the t-test results has been added to S2 Table. We have also included this finding in the Results section.

Results (line 207-210)

Comparison was performed with five normal controls without liver disease and the high-level group among the four variables. Differentially expressed miRNAs between normal and high-level groups are presented (S4 Fig), and the matrix of their expression levels were provided (S1 Table).

S4 Fig. Differentially expressed miRNAs between five normal controls and four patients with high scores. The higher score among the four scores (steatosis score 3; n = 10, inflammation score 3; n = 6, ballooning score 1 or 2; n = 22, and NAS 6-8; n = 10) and five normal controls were compared. (A) A total of 19 miRNAs with a difference in expression between five normal controls and 10 steatosis score 3 (S3; p < 0.001 and |fold change| > 1.8). (B) A total of 17 miRNAs with a difference in expression between five normal controls and six inflammation score 3 (I3; p < 0.001 and |fold change| > 1.5). (C) A total of 17 miRNAs with a difference in expression between five normal controls and 22 ballooning scores of 1 or 2 (p < 0.001 and |fold change| > 1.8). HB indicates a high ballooning score, considered as a score of 1 or 2. (D) A total of 17 miRNAs with a difference in expression between five normal controls and 10 NAS scores of 6- 8 (S678; p < 0.001 and |fold change| > 1.1). (E) Venn diagram and table by variable for each miRNA. S, I, B, and N indicate steatosis, inflammation, ballooning, and NAS (nonalcoholic fatty liver disease (NAFLD) activity score), respectively.

2) Is there any correlation between the miRNAs identified with diabetes, dyslipidemia, or gender?

Response: Thank you for your insightful comment. For the eight miRNAs presented in this study, 24 analyses and visualizations (boxplot) for diabetes, dyslipidemia, and sex were performed. Some miRNAs showed significant (p < 0.05) results; however, most could not explain the above three variables. The results of the analysis are summarized in S6 Fig.

Results (line 291-293)

The eight miRNAs showed little statistical difference according to three variables (diabetes, dyslipidemia, or sex) and can be considered confounding variables (S5 Fig.).

S6 Fig. Boxplot of significance and expression levels according to the confounding variables of eight miRNAs suggested to explain NAFLD in this study. Eight miRNAs (let-7b-5p, miR-378h, -1184, -3613-3p, -877-5p, -602, -133b, and 509-3p) and three confounding variables (sex, diabetes, and dyslipidemia) were listed, and the expression levels are presented as boxplots with significant values (p-values). NAFLD, nonalcoholic fatty liver disease.

3) The reference mentioned in the manuscript for the exosome isolation method is related with exosome isolation from cell cultures. Could the authors describe the procedure of exosome isolation? Including volume of serum used, sample preservation, etc…

Response: Thank you for your insightful comment. We detailed the isolation of exosomes from serum. We have also added sample volume and preservation in the Materials and Methods section.

Materials and Methods (line 107-114, 124-132)

Serum collection and exosomal isolation

After serum collection from patients with NAFLD and controls, sera were stored at -80 °C and thawed just before exosome isolation. We isolated exosomes from sera using an exosome isolation kit (ExoQuick Plus, Systemic Biosciences, CA, USA) according to the manufacturer’s protocol. Sera (1 mL) were centrifuged at 3000 × g for 15 min to remove cell and cell debris. Exoquick solution (63 μL) was added to the sera and incubated for 1 h at 4 °C. Mixtures were centrifuged at 1500 × g for 30 min, and pellets were collected. The exosome pellets were resuspended in 150 µL of phosphate-buffered saline.

RNA extraction and microarray analysis

RNA extraction from exosomes was performed in accordance with a previously published protocol [17]. Microarray analysis was performed using an Affymetrix GeneChip™ miRNA 4.0 array, and image and signal data were extracted using GeneChip™ Scanner 3000DX and Transcriptome Analysis Console 4.0. Normalization of each type of data was performed using the robust multichip average and detection above background algorithms in the Affymetrix Expression Console software. Normalized data are presented as a matrix, converted to an R data frame, and then subjected to statistical analyses. Analyses have been performed with R statistics v.3.6.1.

4) The exosomes isolated from the serum should be assessed by analyzing vesicle diameter by electron microscopy or by assessing common exosomal proteins by western blot.

Response: Thank you for your important comment. We assessed exosomal proteins (CD9 and CD63) by western blotting in S2 Fig. We have included a methodological description for western blotting in the Materials and Methods section.

Materials and Methods (line 116-122)

Western blotting

Exosomal proteins were isolated using the M-PER buffer. Twenty micrograms of exosomes was used for immunoblotting using CD9 and CD63 antibodies (Systemic Biosciences, CA, USA). Protein separation was performed by electrophoresis using 10% sodium dodecyl sulfate-polyacrylamide gels, and the proteins were transferred to nitrocellulose membranes. Protein bands were visualized using enhanced chemiluminescence (Perkin Elmer, Waltham, MA, USA).

5) A better description for the inclusion and exclusion criteria should be included, define alcohol consumption for women and men, viral-, autoimmune-, hemochromatosis- and drug-induced causes of liver disease, etc…

Response: We added detailed inclusion and exclusion criteria in the Materials and Methods section, as described below.

Materials and Methods (line 81-87)

Study population

A total of 41 patients with NAFLD were included in this study. The inclusion criterion was the confirmation of NAFLD with liver biopsy. The exclusion criteria were as follows: 1) patients with other chronic liver diseases, including chronic viral hepatitis B and C and autoimmune liver disease, 2) excessive alcoholic consumption (men > 30 g/day and women > 20 g/day), 3) patients with decompensated liver cirrhosis, 4) patients with other severe systemic disease, and 5) pregnant women.

6) Can the authors be more specific with the classification used to classify the patients into groups with low, medium, and high NAS scores.

Response: As shown in the patient information in Table 1, the NAS scores (2/3/4/5/6/7/8) of the patients enrolled in this study were 7 (17.1%)/8(19.5%)/9 (22%)/8 (19.5%)/6 (14.6%)/2 (4.9%)/1 (2.4%). Classification into four groups was considered when designing the experiment; however, statistical significance could not be obtained because the number of patients included in one group decreased. In this study, 41 patients were enrolled, and a more detailed classification was difficult and will be performed in a future study.

7) The supplementary figure 3 contain important information about the miR-RNAs identified, it would be useful to move this supplementary figure to main figures.

Response: Thank you for your insightful comments. We revised the supplementary figure3 as Figure 2.

8) It should be also helpful to discuss the known functions of some of the MiR-RNAs identified, as for instance miR-125b-5p (PMID: 26924666), miR12245p (PMID: 30087537), miR-133b etc.

Response: We have mentioned in the discussion about miRNAs and mRNAs from the interaction study. Although it was difficult to find genes directly related to liver injury, the miRNA-mRNA interaction presented in this study will serve as evidence to explain NAFLD and NASH well.

Discussion (line 342-346, 364-368)

To date, differentially expressed miRNAs in liver injury status, such as oxidative stress or viral infection, have been detected [21, 26-28]. It is necessary to comprehensively understand the relationship between miRNA and liver enzymes derived from biochemical analyses, as well as miRNA and mRNA derived from previous studies related to the liver injury status.

We attempted to link the miRNAs and mRNAs that have been previously studied in pathological liver injury such as NAFLD, NASH, and liver fibrosis [28, 34-36]. Although it was difficult to identify genes directly related to liver injury in the interaction study, the miRNA-mRNA interaction presented in this study will serve as evidence to explain NAFLD and NASH well.

---

## [Editor Report · Decision Letter 1]

26 Jul 2021

Evaluation of the severity of nonalcoholic fatty liver disease by the analysis of serum exosomal miRNA expression

PONE-D-21-16287R1

Dear Dr. Yeon,

We’re pleased to inform you that your manuscript has been judged scientifically suitable for publication and will be formally accepted for publication once it meets all outstanding technical requirements.

Kind regards,

Matias A Avila, Ph.D.

Academic Editor

PLOS ONE
---

## [Editor Report · Acceptance letter]

28 Jul 2021

PONE-D-21-16287R1 

Evaluation of the severity of nonalcoholic fatty liver disease through analysis of serum exosomal miRNA expression 

Dear Dr. Yeon:

I'm pleased to inform you that your manuscript has been deemed suitable for publication in PLOS ONE. Congratulations! Your manuscript is now with our production department. 

Kind regards, 

on behalf of

Dr Matias A Avila 

Academic Editor

PLOS ONE